# Robust Moiety Model Selection Using Mass Spectrometry Measured Isotopologues

**DOI:** 10.3390/metabo10030118

**Published:** 2020-03-21

**Authors:** Huan Jin, Hunter N.B. Moseley

**Affiliations:** 1Department of Toxicology and Cancer Biology, University of Kentucky, Lexington, KY 40536, USA; huan.jin@uky.edu; 2Department of Molecular & Cellular Biochemistry, University of Kentucky, Lexington, KY 40536, USA; 3Markey Cancer Center, University of Kentucky, Lexington, KY 40536, USA; 4Resource Center for Stable Isotope Resolved Metabolomics, University of Kentucky, Lexington, KY 40536, USA; 5Institute for Biomedical Informatics, University of Kentucky, Lexington, KY 40536, USA

**Keywords:** stable isotope resolved metabolomics (SIRM), moiety modeling, model selection, isotopologue deconvolution, overfitting, nonlinear inverse problem

## Abstract

Stable isotope resolved metabolomics (SIRM) experiments use stable isotope tracers to provide superior metabolomics datasets for metabolic flux analysis and metabolic modeling. Since assumptions of model correctness can seriously compromise interpretation of metabolic flux results, we have developed a metabolic modeling software package specifically designed for moiety model comparison and selection based on the metabolomics data provided. Here, we tested the effectiveness of model selection with two time-series mass spectrometry (MS) isotopologue datasets for uridine diphosphate N-acetyl-d-glucosamine (UDP-GlcNAc) generated from different platforms utilizing direct infusion nanoelectrospray and liquid chromatography. Analysis results demonstrate the robustness of our model selection methods by the successful selection of the optimal model from over 40 models provided. Moreover, the effects of specific optimization methods, degree of optimization, selection criteria, and specific objective functions on model selection are illustrated. Overall, these results indicate that over-optimization can lead to model selection failure, but combining multiple datasets can help control this overfitting effect. The implication is that SIRM datasets in public repositories of reasonable quality can be combined with newly acquired datasets to improve model selection. Furthermore, curation efforts of public metabolomics repositories to maintain high data quality could have a huge impact on future metabolic modeling efforts.

## 1. Introduction

While the first observations of metabolic alterations in cancer were made about a century ago [1], metabolomics is a relatively new field of ‘omics’ technology aiming to systematically characterize metabolites being created and/or utilized in cells, tissues, organisms, and ecosystems [2]. This combined consumption and biosynthesis of metabolites can be represented as flux through specific metabolic paths within cellular metabolism, reflecting specific physiological and pathological states in biomedically useful detail and in ways that are distinct and often more sensitive than other omics methods. It is increasingly recognized that metabolomics biomarkers have great utility in characterizing and monitoring diseases with significant metabolic reprogramming like cancer [3]. Therefore, better regulatory understanding of specific metabolic flux phenotypes of metabolic diseases will aid in developing new therapeutic strategies.

Stable isotope resolved metabolomics (SIRM) experiments utilize stable isotopes from a labeling source to isotopically enrich detected metabolite analytical features, providing more complex but data-rich metabolomics datasets for metabolic flux analysis. Advances in mass spectrometry (MS) and nuclear magnetic resonance spectroscopy (NMR) greatly contribute to the generation of high-quality SIRM datasets [4]. However, computational methods are required to gain biologically meaningful interpretation from such complex datasets, especially in terms of metabolic flux through specific metabolic paths in cellular metabolism. Most current metabolic flux analysis methods heavily depend on a predetermined metabolic network and are mostly focused on the analysis of ^13^C tracer experiments [5,6,7,8]. However, large numbers of ‘unknown’ metabolites in the metabolomics datasets strongly indicate that current metabolic networks are far from complete, especially for secondary metabolism and central metabolism of non-model organisms [9,10,11]. Without an accurate and reasonably defined metabolic network, it is challenging to conduct meaningful metabolic flux analyses. Even worse, assuming that a metabolic model is accurate compromises the scientific rigor of the metabolic modeling and can lead to misinterpretation of results [12].

Our newly developed moiety deconvolution package called moiety_modeling is a novel method for analyzing time series SIRM MS isotopologue profiles that can involve single or multiple isotope tracers [13]. This package integrates facilities for moiety (i.e., biochemical functional group) model and data representation, model (parameter) optimization, analysis of optimization results, and model selection under a single moiety modeling framework. A typical data analysis workflow for this moiety modeling framework is shown in Figure 1. Moiety modeling deconvolutes isotopologue intensity data of a metabolite into pseudo-isotopomers based on a given moiety description of the metabolite. Moiety modeling is an early step in certain metabolic flux analysis approaches that can allow the comparison of different moiety models for model selection. First, plausible and hypothetical moiety models of an interesting metabolite are provided by a user based on a relevant metabolic network. After the optimization of each moiety model during isotopologue deconvolution, the best model provided can be selected based on the optimized results of model parameters, which can be directly used for downstream metabolic flux analysis and interpretation.

In this paper, we used this moiety modeling framework to investigate the effects of the optimization method, optimizing degree, objective function, and selection criterion on model selection to identify modeling criteria that promote robust model selection. To our knowledge, this is the first attempt to investigate how all of these factors can affect model selection in metabolic modeling.

## 2. Results

### 2.1. UDP-GlcNAc Moiety Model Construction

UDP-GlcNAc can be divided into four distinct moieties: glucose, ribose, acetyl, and uracil, in which isotopes incorporate through a metabolic network from an isotope labeling source. The expected (expert-derived) moiety model of ^13^C isotope incorporation from ^13^C-labeled glucose to UDP-GlcNAc (see Figure 2B) is built based on well studied human central metabolism pathways that converge in UDP-GlcNAc biosynthesis, which is corroborated with NMR data [14]. This expert-derived model is labeled as 6_G1R1A1U3, representing six optimizable parameters, one for the glucose moiety (G1), one for the ribose moiety (R1), one for the acetyl moiety (A1), and 3 for the uracil moiety (U3), for each moiety state equation representing the fractional ^13^C incorporation for each moiety. For example, the g6 state represents the incorporation of ^13^C_6_ into the glucose moiety, whereas the g0 state represents no incorporation of ^13^C. Since both g0 and g6 must sum to 1, there is only one parameter that needs to be optimized for this moiety state equation. The set of isotopologue intensity equations are derived using the moiety model parameters and Equation (1), as illustrated for the expert-derived model in Figure 2B. Figure 2C shows an alternative hypothetical moiety model 7_G0R3A1U3_g3R2R3_g6r5_r4 along with the isotopologue intensity equations generated from the model.
(1)$$I_{x,calc}=\underset{ic_{a}\in IC_{x}}{\sum }ic_{a}\;;\;IC_{x}=\left\{ic_{v}|isotope\_content\left(ic_{v}\right)=x\right\}\;;\;ic_{v}=\underset{j}{\prod }moiety\_state_{j,v_{j}}\;$$

We also manually crafted 40 hypothetical moiety models to capture isotope flow from [U-^13^C]-glucose into each moiety. This set of models provides a mechanism for testing how robustly the expert-derived model can be selected from all the other models provided.

### 2.2. A Simple Comparison of Two Moiety Models

Model optimization aims to minimize an objective function that compares calculated isotopologues based on moiety state parameters from the model to the directly observed, experimentally-derived isotopologues. Figure 3A shows the comparison of optimized model parameters between the expert-derived moiety model (6_G1R1A1U3) and the hypothetical moiety model (7_G0R3A1U3_g3r2r3_r4) for three time points of isotopologue intensity data, i.e., three sets of isotopologue intensities. In these model optimizations, the SAGA-optimize method and absolute difference objective function were used and DS0, DS1, and DS2 correspond to the 34 h, 48 h, and 72 h time points in the FT-ICR-MS UDP-GlcNAc dataset. We can easily tell that the relative intensity of the corresponding model parameters between these two models are quite different, suggesting that the moiety-specific ^13^C isotopic incorporation derived from the same MS isotopologue profile varies from one model to another. Furthermore, experiment-derived and model parameter-calculated isotopologue profiles are shown in Figure 3B, illustrating how much better the expert-derived model vs. an inaccurate model is able to reflect the observed data.

### 2.3. Effects of Optimization Method on Model Selection

The first question we were interested in was whether the optimization method could affect the model selection results. As in the previous analysis, we used three time points from the FT-ICR-MS dataset, the AICc criterion, and an absolute difference objective function in the initial trial. The optimization for each model was conducted 100 times, and we used the average of the 100 optimization results in the analysis (see Table 1). Most optimization methods can select the expert-derived model except for ‘SLSQP’. What interested us most was that the ‘SLSQP’ method failed in model selection with the lowest loss value (value returned from objective function) and is generally considered to be the fastest converging of the optimization methods we tested.

We repeated the experiment with the ‘SLSQP’ method 10 times and found that model selection fails when the loss value approaches 0.3 (Appendix A), suggesting strong instability of model selection at a critical point. Model optimization aims to minimize the objective function, which is actually a non-linear inverse problem and one inherited issue in solving a non-linear inverse problem is overfitting (i.e., fitting to error in the data). Therefore, we developed the hypothesis that over-optimization of model parameters can lead to failure in model selection.

### 2.4. Over-optimization Leads to Failure in Model Selection

To test the above hypothesis, we first tried to increase the stop criterion of the ‘SLSQP’ method to control over-optimization. The results are shown in Table 2. When optimization stops earlier, the expert-derived model can be selected, which supports our hypothesis.

The SAGA-optimize method is more flexible in controlling the degree of optimization simply by adjusting the number of optimization steps. The more steps, the lower the average loss value reached by the optimization. Next, we performed a set of experiments using the SAGA-optimize method with increasing number of optimization steps to further validate the hypothesis. The results are summarized in Table 3. We can see that the loss value decreases as optimization step increases. When the loss value reaches a certain critical point, the expert-derived model cannot be selected, further supporting the hypothesis that over-optimization can lead to failure in model selection. Furthermore, the selected model can change with increasing degrees of over-optimization.

Based on the above results, we conclude that it is not the optimization method but the degree of optimization that affects model selection, which is explained by overfitting to error in the data when solving a non-linear inverse problem. When optimization reaches a certain critical point, successful model selection cannot be guaranteed. Therefore, proper control of the degree of optimization is of great importance in model selection.

### 2.5. Effects of Selection Criterion on Model Selection

Next, we investigated whether selection criterion could affect model selection. We compared the model selection results generated by SAGA-optimize with different model selection criteria (see Table 4, see Appendix A for complete results). From these results, we can see that the rank of top models is quite consistent across different selection criteria, suggesting that these model selection criteria have little effect on robust model selection, at least under this model selection context. Since our previous experiments used AICc as the selection criterion, we will stick with AICc in the following experiments.

### 2.6. Effects of Selection Criterion on Model Selection

Considering that the dominant type of error existing in metabolomics datasets may vary from dataset to dataset, different forms of an objective function may affect model optimization and then influence the results of model selection. Here, we tested the effects of four objective functions in the context of model selection: absolute difference, absolute difference of logs, square difference, and difference of AIC. To speed up optimization, we first split the FT-ICR-MS dataset based on time point (34 h, 48 h, 72 h) into separate model optimizations executed on their own CPU core, and then combine the optimization results for the model selection. This functionality is provided by the moiety_modeling package. We set a series of experiments for each objective function with the SAGA-optimize method. The results are shown in Table 5, Table 6, Table 7 and Table 8. In comparing Table 5 to Table 3, the number of optimizations per time point provides roughly the same degree of optimization as three times the number of optimization steps used on a combined optimization.

From these tables, we can see that optimization with the absolute difference of logs objective function is less likely to fail (>250,000 steps) in the model selection compared to the other three objective functions (10,000–20,000 steps). One interpretation from these results is that the FT-ICR-MS dataset is dominated by proportional error instead of additive error. However, the AICc produced with the absolute difference of logs is significantly higher (less negative) than that produced by the other objective functions. Therefore, this objective function may simply be hindering efficient optimization, especially if the dataset is dominated by an error structure that is not as compatible with this objective function. From this alternative viewpoint, additive error may actually dominate this dataset. We used a graphical method to visualize errors in both FT-ICR-MS and LC-MS datasets (Figure 4 and Figure 5). For the plots of FT-ICR-MS datasets, we used another dataset generated from the same procedure, which included two replicates at 0, 3 h, 6 h, 11 h, 24 h, 34 h, and 48 h time points. For two replicates with proportional error, a scatter plot of each replicate against the other will show an increasing spread of values with increasing signal and the log-transformed data will collapse into a line. Plot of two replicates with additive error can be viewed as uniformly deviated from the line of identity, but once log-transformed will show an increasing spread of values with decreasing signal. The original plots of raw data indicate existence of proportional error in both FT-ICR-MS and LC-MS datasets (Figure 4A and Figure 5A). However, the original plots of normalized data almost collapsed to a straight line (Figure 4C and Figure 5C), suggesting that normalization somehow removes the proportional error in the raw data. In addition, from the log-transformed plots, we can see that additive error does not exist in the normalized FT-ICR-MS datasets (Figure 4D) but does exist in the normalized LC-MS datasets (Figure 5D). The replicate plots of all time points (Appendix A) show similar tendency with selected optimized datasets. Based on the above results, the absolute difference of logs objective function can hinder efficient optimization in FT-ICR-MS datasets. We also compared four objective functions in the context of model selection with LC-MS datasets (Appendix A). From these tables, we can see that model selection fails earlier with absolute difference of logs objective function compared to other objective functions, also suggesting that additive error may dominate in the normalized LC-MS datasets. Based on the above results, the objective function clearly affects model selection and the selection of certain objective functions for model optimization is able to increase resistance to failure in model selection caused by over-optimization; however, this is likely due to less efficient model optimization caused by the selection of an objective function not appropriate for the type of error in the data. 

### 2.7. Effects of Information Quantity on Model Selection

From the above experiments, we found that over-optimization is a primary cause for failure in model selection and this is affected by the objective function used. The next question is whether the quantity of information affects model selection. One basic approach is to utilize more datasets in order to overcome the effects of over-optimization. In the following experiments, we repeated single model optimization 10 times in order to pragmatically finish these computational experiments. Every experiment was conducted 10 times using the AICc criterion and the absolute difference objective function. We used the SAGA-optimize method to test where model selection starts to fail.

First, we used decreasing number of time points of the LC-MS dataset to test whether data quantity affects model selection (Figure 6A, Appendix A). However, model selection failed with few optimization steps when all five time points were included and when only one time point was included, with the most robust model selection occurring with 3 time points. Initially, these results were not expected, until we realized that the relative isotopologue intensity of the 0 and 6 h time points is concentrated within the ^13^C_0_ isotopologue with zero ^13^C tracer. Thus, these datasets are less informative with respect to capturing the isotope flow from labeling source to each moiety in the metabolite. When the 0 and 6 h time points are removed, the selection results improved significantly. Likewise, when information-rich time points are removed, the model selection robustness decreases as well. Similar results were obtained when testing the FT-ICR-MS dataset (Figure 6B). Taken together, the addition of information-rich data contributes to successful model selection while the addition of information-poor data detracts from successful model selection.

To further test this concept, we investigated whether combining FT-ICR-MS (34 h, 48 h, 72 h) and LC-MS (12 h, 24 h, 36 h) datasets can prevent failure in model selection (Figure 6C). From the comparison, we can see that combining information-rich FT-ICR-MS and LC-MS datasets is much more resistant to failure of model selection than just using the information-rich FT-ICR-MS or LC-MS dataset, strongly supporting our previous conclusions that utilizing more information-rich datasets can prevent failure in model selection. Similar results were obtained with absolute difference of logs objective function (Appendix A). These datasets were collected at different times, on very different mass spectrometry platforms. One used chromatographic separation while the other utilized direct infusion. However, the really surprising part is that the datasets were derived from different human cell cultures: LnCaP-LN3 human prostate cancer cells and human umbilical vein endothelial cells.

## 3. Discussion

Here, we discuss the importance of model selection in isotopic flux analysis as a proxy for metabolic flux analysis and factors that affect robust model selection. We found that it is not the optimization method per se, but the degree of optimization that influences model selection, due to the effects of over-optimization, i.e., fitting of model parameters based on the error in the data. Overfitting is a known problem typically due to the ill-conditioning of the nonlinear inverse problem that is partially ill-posed. Moreover, the objective function in model optimization is also of great importance in model selection. Proper selection of an objective function can help increase resistance to failure in model selection. This may mean that different objective functions should be used for model selection versus parameter optimization for flux interpretation. Most SIRM experimental datasets have few collected replicates and time points due to the cost and effort required to acquire these datasets. The lack of replicates makes it impractical to directly estimate error in many of these datasets. Moreover, the presence of different types of systematic error like ion suppression can limit the overall effectiveness of replicate-based error analysis. With our moiety modeling framework, we are able to conduct a set of gradient experiments with varying amounts of optimization (i.e, number of optimization steps) using the SAGA-optimize method to estimate the failure point in model selection caused by overfitting. Furthermore, we found that incorporation of less informative datasets can hinder successful model selection, since they lack appreciable signal representing the incorporation of isotopes simulated by moiety models but with the full amount of error of the measured isotopologues. This lowers the overall isotope incorporation signal to noise ratio, which can lead to increased error in model selection. On the other hand, combining informative datasets (i.e., time points with significant isotope incorporation) can help control failure in model selection, which suggests that informative datasets in public metabolomics repositories can be combined to facilitate robust model selection. Moreover, these datasets do not need to come from identical biological systems, just biological systems that utilize the same part of metabolism being measured and modeled. The implication is that SIRM datasets in public repositories of reasonable quality can be combined with newly acquired datasets to improve model selection. Furthermore, curation efforts of public metabolomics repositories to maintain high data quality and provide metrics of measurement error could have a huge impact on future metabolic modeling efforts.

## 4. Materials and Methods

### 4.1. UDP-GlcNAc Time Course MS Isotopologue Datasets

Two UDP-GlcNAc time course MS isotopologue datasets were used to test the robustness of model selection mechanism. The first is a direct infusion Fourier transform MS (FTMS) UDPGlcNAc ^13^C isotopologue dataset derived from LnCaP-LN3 human prostate cancer cells with [U-^13^C]-glucose as isotope labeling source and collected on an Advion Nanomate nanoelectrospray inline connected to a Thermo 7T LTQ Fourier transform ion cyclotron resonance MS (FT-ICR-MS). This dataset includes 3 time points: 34 h, 48 h, and 72 h [14]. The second is a liquid chromatography-MS (LC-MS) UDP-GlcNAc ^13^C isotopologue dataset derived from human umbilical vein endothelial cells with [U-^13^C]-glucose as the isotope labeling source and collected on a ThermoFisher Dionex UltiMate 3000 LC System in-line connected to a ThermoFisher Q-Exactive Orbitrap MS. This dataset has 5 time points: 0 h, 6 h, 12 h, 24 h and 36 h [15]. Look to the reference associated with each dataset for more details on their experimental design, implementation, and rationale.

### 4.2. Objective Functions

We used four distinct forms of the objective function (Table 9) that compares the observed isotopologues and corresponding calculated isotopologues derived from model parameters obtained from model optimization. The first is a summation of absolute differences between observed and calculated isotopologues, which is generally expected to work well with data where the dominant type of error is additive. The second is a summation of the absolute differences between the log of observed and calculated isotopologues, which is generally expected to work well with data where the dominant type of error is proportional. The third is a summation of square of differences between observed and calculated isotopologues. The fourth one tries to mimic the effect of model selection criteria. 

### 4.3. Optimization Methods

From a mathematics perspective, model optimization is actually a non-linear inverse problem. Several different optimization methods were used to solve this problem, including the SAGA-optimize method [13], and three other optimization methods (‘TNC’ [16], ‘SLSQP’ [17], and ‘L-BFGS-B’ [18]) available in the scipy.optimize Python module. The SAGA-optimize method is a combination of simulated annealing and genetic algorithm optimization methods that utilizes the population and crossover concepts from genetic algorithm to improve the optimization speed and consistency over older more traditional implementations of simulated annealing, allowing SAGA-optimize to produce better quality optimization results more efficiently (i.e., with fewer overall number of optimization steps). The ‘TNC’ method is designed for optimizing non-linear functions with large numbers of independent variables [16]. The SLSQP method uses Sequential Least Squares Programming, which is an iterative method for constrained nonlinear optimization [17]. ‘L-BFGS-B’ is a limited-memory algorithm for solving large nonlinear optimization problems subject to simple bounds on the variables [18].

### 4.4. Model Selection Estimators

We used three different quality estimators (Table 10) in model selection: the Akaike Information Criterion (AIC) [19], the sample size corrected Akaike Information Criterion (AICc) [20], and the Bayesian Information Criterion (BIC) [21]. The Akaike information criterion (AIC) is biased to select models with more parameters when the sample size is small, which can lead to overfitting [19]. The sample size corrected AIC (AICc) was developed to handle this bias and prevent overfitting [20]. The Bayesian information criterion (BIC) is another criterion commonly used in model selection [21].

### 4.5. Computer Code and Software

The moiety_modeling and SAGA-optimize packages are available on: GitHub: https://github.com/MoseleyBioinformaticsLab/moiety_modeling, https://github.com/MoseleyBioinformaticsLab/SAGA_optimize. PyPI: https://pypi.org/project/moiety-modeling/, https://pypi.org/project/SAGA-optimize/.

Project documentation is available online at ReadTheDocs: https://moiety-modeling.readthedocs.io/en/latest/, https://saga-optimize.readthedocs.io/en/latest/.

## Figures and Tables

**Figure 1 metabolites-10-00118-f001:**
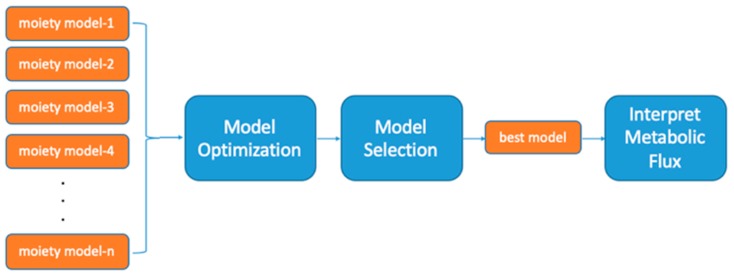
Workflow for Moiety Modeling.

**Figure 2 metabolites-10-00118-f002:**
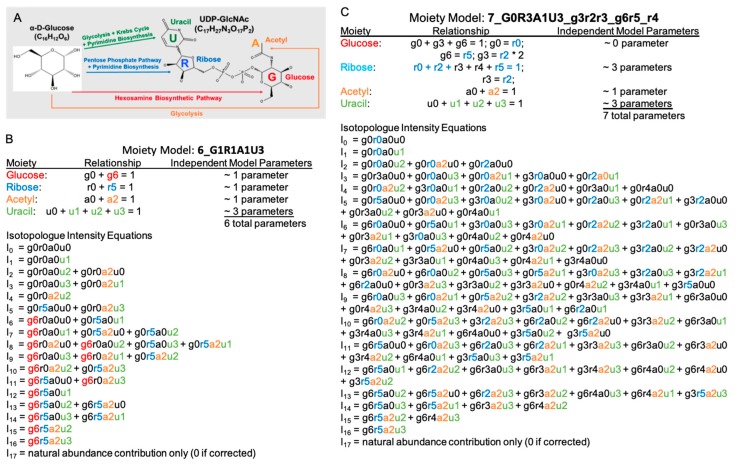
Example complex metabolite UDP-GlcNAc and associated moiety models. (**A**) Major human metabolic pathways from glucose to the four moieties of UDP-GlcNAc. (**B**) The expert-derived moiety model based on known human central metabolism pathways with corroborating NMR data. (**C**) An alternative hypothetical moiety model with simple perturbations of the original expert-derived model.

**Figure 3 metabolites-10-00118-f003:**
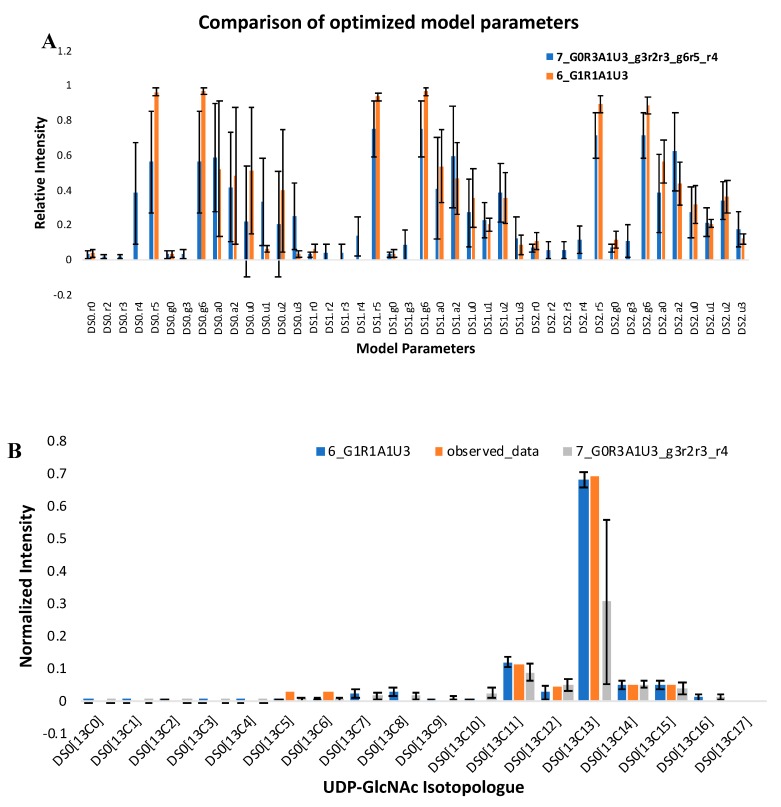
Optimized results for 6_G1R1A1U3 and 7_G0R3A1U3_g3r2r3_r4 models. Each model optimization was conducted 100 times. (**A**) Comparison of mean of optimized model parameters with standard deviation. (**B–D**) Reconstruction of the isotopologue distribution of UDP-GlcNAc from model parameters. Observed isotopologue data were compared with the mean of calculated isotoplogue data with standard deviation from the optimized parameters for each model.

**Figure 4 metabolites-10-00118-f004:**
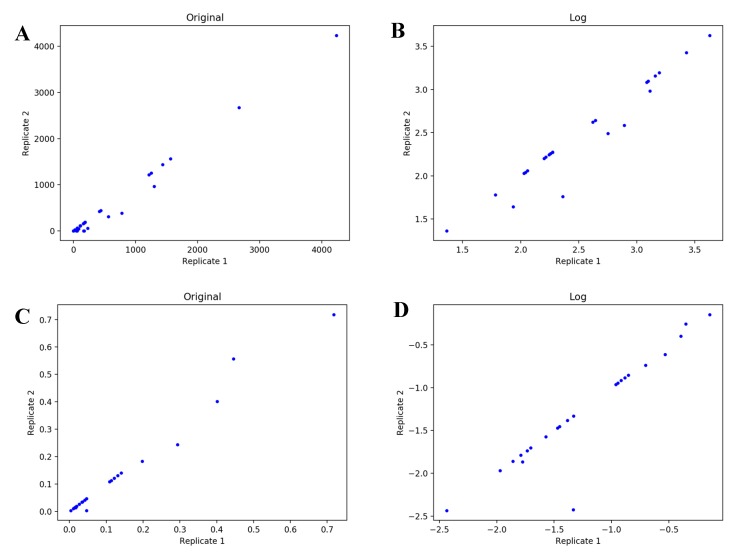
Error analysis in FT-ICR-MS datasets. (**A**,**B**) are plots of raw data. (**C**,**D**) are plots of renormalized data after natural abundance correction. All these plots contain 3 time points (24–48 h).

**Figure 5 metabolites-10-00118-f005:**
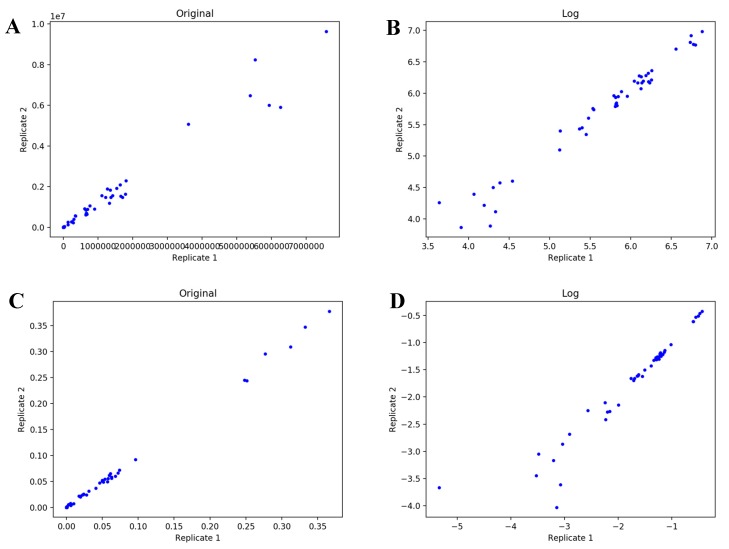
Error analysis in LC-MS datasets. (**A**,**B**) are plots of raw data. (**C**,**D**) are plots of renormalized data after natural abundance correction. All these plots contain 3 time points (12–36 h).

**Figure 6 metabolites-10-00118-f006:**
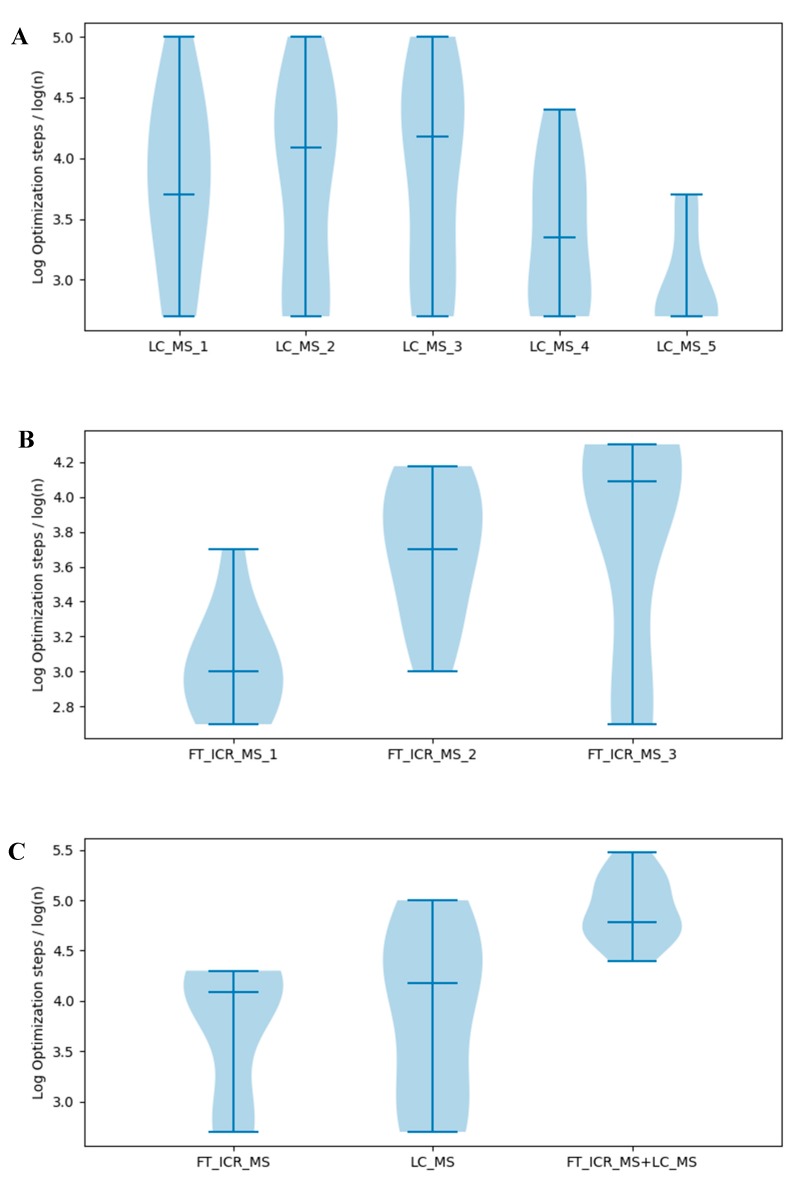
Comparison of the log optimization steps where model selection with different datasets begins to fail. (**A**) Test with LC-MS datasets. LC-MS_1 to LC-MS_5 represent LC-MS datasets with 36 h, 24–36 h, 12–36 h, 6–36 h and 0–36 h. (**B**) Test with FT-ICR-MS datasets. FT-ICR-MS_1 to FT-ICR-MS_3 represent FT-ICR-MS datasets with 48 h, 48–72 h and 34–72 h. (**C**) Test with combination of LC-MS (12 h, 24 h, 36 h) and FT-ICR-MS (34 h, 48 h, 72 h) datasets. The median values are indicated in the plots.

**Table 1 metabolites-10-00118-t001:** Comparison of optimization methods in model selection.

Optimization Method	Loss Value	AICc	Selected Model
SAGA	0.469	−401.760	Expert-derived model
**SLSQP**	**0.320**	**−408.341**	**7_G2R1A1U3_g5**
L-BFGS-B	0.763	−342.164	Expert-derived model
TNC	0.870	−327.344	Expert-derived model

Dataset: FT-ICR-MS (combined); Selection criterion: AICc; Objective function: Absolute difference. Bolded line indicates incorrect selection of the expert-derived model.

**Table 2 metabolites-10-00118-t002:** Over optimization experiments with the ‘SLSQP’ method.

Optimization Method	Loss Value	AICc	Selected Model	Stop Criterion
SLSQP	0.320	−408.341	7_G2R1A1U3_g5	‘ftol’: *1e-06*
SLSQP	0.514	−393.934	Expert-derived model	‘ftol’: *1e-05*

Dataset: FT-ICR-MS (combined); Selection criterion: AICc; Objective function: Absolute difference.

**Table 3 metabolites-10-00118-t003:** Over optimization experiments with SAGA-optimize method.

Optimization Steps	Loss Value	AICc	Selected Model
500	2.070	−219.488	Expert-derived model
1000	1.754	−235.728	Expert-derived model
2000	1.377	−260.654	Expert-derived model
5000	0.941	−305.651	Expert-derived model
10000	0.664	−375.192	Expert-derived model
25000	0.469	−401.760	Expert-derived model
50000	0.408	−414.737	Expert-derived model
**75000**	**0.328**	**−418.228**	**7_G2R1A1U3_g5**
100000	0.316	−424.924	7_G1R2A1U3_r4

Dataset: FT-ICR-MS (combined); Selection criterion: AICc; Objective function: Absolute difference. Bolded line indicates the start of incorrect selection of the expert-derived model.

**Table 4 metabolites-10-00118-t004:** Comparison of mode rank based on different model selection criteria.

Models	AICc	Rank	AIC	Rank	BIC	Rank
**Expert-derived model**	**−401.7597**	**1**	**−421.3026**	**1**	**−385.5009**	**1**
7_G1R1A2U3	−384.3075	2	−413.1825	2	−371.4139	2
7_G2R1A1U3_g5	−381.2868	3	−410.1618	3	−368.3932	3
7_G1R2A1U3_r3	−379.2657	4	−408.1407	4	−366.3720	4
7_G1R2A1U3_r4	−378.8969	5	−407.7719	5	−366.0033	5
7_G2R1A1U3_g4	−375.9538	6	−404.8288	6	−363.0601	6
9_G2R2A2U3_r2r3_g5	−254.4087	37	−312.5625	36	−258.8599	37
9_G2R2A2U3_r2r3_g3	−248.2277	38	−306.3815	38	−252.6789	38
9_G2R2A2U3_r2r3_g2	−242.9984	39	−301.1522	39	−247.4497	39
9_G2R2A2U3_r2r3_g1	−242.4110	40	−300.5648	40	−246.8623	40
7_G0R3A1U3_g3r2r3_g6r5_r4	−226.7271	41	−255.6021	41	−213.8334	41

Dataset: FT-ICR-MS (combined); Optimization method: SAGA-optimize (25,000 steps); Objective function: Absolute difference. Bolded line indicates the start of incorrect selection of the expert-derived model.

**Table 5 metabolites-10-00118-t005:** Model selection test with absolute difference objective function.

Optimization Steps	Loss Value	AICc	Selected Model
500	1.045	−293.540	Expert-derived model
1000	0.819	−330.411	Expert-derived model
2000	0.651	−361.038	Expert-derived model
5000	0.459	−408.167	Expert-derived model
10000	0.392	−422.516	Expert-derived model
15000	0.359	−431.276	Expert-derived model
**20000**	**0.290**	**−434.468**	**7_G1R1A2U3**
25000	0.285	−436.909	7_G1R1A2U3

Dataset: FT-ICR-MS (split); Selection criterion: AICc; Objective function: absolute difference; Bolded line indicates the start of incorrect selection of the expert-derived model.

**Table 6 metabolites-10-00118-t006:** Model selection test with square difference objective function.

Optimization Steps	Loss Value	AICc	Selected Model
500	0.085	−298.516	Expert-derived model
1000	0.047	−330.096	Expert-derived model
2000	0.023	−367.279	Expert-derived model
5000	0.011	−404.509	Expert-derived model
10000	0.007	−425.695	Expert-derived model
**15000**	**0.005**	**−429.869**	**7_G2R1A1U3_g5**
20000	0.005	−435.348	7_G1R2A1U3_r4

Dataset: FT-ICR-MS (split); Selection criterion: AICc; Objective function: square difference. Bolded line indicates the start of incorrect selection of the expert-derived model.

**Table 7 metabolites-10-00118-t007:** Model selection test with difference of AIC objective function.

Optimization Steps	Loss Value	Selected Model
500	−345.559	Expert-derived model
1000	−371.852	Expert-derived model
2000	−398.570	Expert-derived model
5000	−436.582	Expert-derived model
**10000**	**−458.064**	**7_G1R1A2U3**
15000	−467.960	7_G2R1A1U3_g5

Dataset: FT-ICR-MS (split); Selection criterion: AICc; Objective function: difference of AIC. Bolded line indicates the start of incorrect selection of the expert-derived model.

**Table 8 metabolites-10-00118-t008:** Model selection test with absolute difference of logs objective function.

Optimization Steps	Loss Value	AICc	Selected Model
500	31.647	−221.501	Expert-derived model
1000	29.628	−223.363	Expert-derived model
2000	28.164	−224.330	Expert-derived model
5000	27.096	−225.911	Expert-derived model
10000	26.631	−227.499	Expert-derived model
15000	26.469	−227.690	Expert-derived model
20000	26.398	−227.780	Expert-derived model
25000	26.271	−228.178	Expert-derived model
50000	26.126	−228.892	Expert-derived model
100000	25.949	−228.926	Expert-derived model
150000	25.865	−229.926	Expert-derived model
250000	25.777	−230.232	Expert-derived model

Dataset: FT-ICR-MS (split); Selection criterion: AICc; Objective function: absolute difference of logs.

**Table 9 metabolites-10-00118-t009:** Objective functions.

Objective Function	Equation
Absolute difference	Σ|I_n,obs_ – I_n,calc_|
Absolute difference of logs	Σ|log(I_n,obs_) – log(I_n,calc_)|
Square difference	Σ(I_n,obs_ – I_n,calc_)^2^
Difference of AIC	2k+nln(RSS/n)

k is the number of parameters; n is the number of data points; RSS is the residual sum of squares: RSS = $\sum_{i=1}^{n}{\left(I_{obs,i}-I_{calc,i}\right)}^{2}$.

**Table 10 metabolites-10-00118-t010:** Model selection estimators.

Selection Criterion	Equation
Akaike Information Criterion (AIC)	2k+nln(RSS/n)
Sample size corrected AIC (AICc)	$\mathrm{AIC}+\left(2k^{2}+2k\right)/\left(n-k-1\right)$
Bayesian Information Criterion (BIC)	nln(RSS/n)+ kln(n)

k is the number of parameters; n is the number of data points; RSS is the residual sum of squares: RSS = $\sum_{i=1}^{n}{\left(I_{obs,i}-I_{calc,i}\right)}^{2}$.

## Data Availability

All data used and the results generated in this manuscript are available on figshare: https://figshare.com/articles/moiety_model_selection/10279688.

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
