# Peer review of "Robust Moiety Model Selection Using Mass Spectrometry Measured Isotopologues"

_metabolites, 2020, doi:10.3390/metabo10030118_

Round 1

Reviewer 1 Report

  • Please take care of spelling/grammar issues and review the entire manuscript, for example, lines 58-59: “current metabolic networks”
  • Figure 2: Please simplify by removing the majority of the equation. This is a figure and the list does not need to be comprehensive, it needs to be explanatory. Instead of actual equations for each of the moiety model, can colored dots be used?
  • When you are showing the transition from the Expert-derived model to other model, for example in Table 3, could you show more data points (i.e., optimization steps)? A transition like this calls for a higher resolution of the shift. So instead of 50000 to 75000, could you show more steps so that it is easier to grasp where exactly the shift happens?
  • Move table 4 to Supplementary.
  • Your observation of model selection failed with few optimization steps when “all data points were included” and “only one data point was included seems interesting (lines 266-269). Could you elaborate on your discussion on this observation? How would someone, without prior knowledge of the distribution of specific isotopes, decide how many, and which data points to consider? What should be an appropriate pre-examination of data to help in this decision making?
  • Please elaborate on section 4.3 (Optimization methods) for a more general audience.
  • Why are there two sections named “Optimization methods”?

Author Response

Reviewer 1:

Issue 1:

Please take care of spelling/grammar issues and review the entire manuscript, for example, lines 58-59: “current metabolic networks”

Response:

We have reviewed the entire manuscript again and have tried to remove all grammatical mistakes.

Issue 2:

Figure 2: Please simplify by removing the majority of the equation. This is a figure and the list does not need to be comprehensive, it needs to be explanatory. Instead of actual equations for each of the moiety model, can colored dots be used?

Response:

This is a matter of preference of the reader.  The inclusion of the equations shows exactly what is used in the expert-derived moiety model.  We have had other reviewers demand that we include these equations for completeness in prior publications.  But we can understand that some readers may not be interested in this level of detail.  However, we would error in providing a level of detail that is interesting to some readers, but can be simply ignored by other readers.

Issue 3:

When you are showing the transition from the Expert-derived model to other model, for example in Table 3, could you show more data points (i.e., optimization steps)? A transition like this calls for a higher resolution of the shift. So instead of 50000 to 75000, could you show more steps so that it is easier to grasp where exactly the shift happens?

Response:

We have added the additional optimization steps into versions of tables 3,5,6, and 7 in the supplemental material.

Issue 4:

Move table 4 to Supplementary.

Response:

We have truncated Table 4 and provided a full table in supplemental material.

Issue 5:

Your observation of model selection failed with few optimization steps when “all data points were included” and “only one data point was included seems interesting (lines 266-269). Could you elaborate on your discussion on this observation? How would someone, without prior knowledge of the distribution of specific isotopes, decide how many, and which data points to consider? What should be an appropriate pre-examination of data to help in this decision making?

Response:

The inclusion of time points with limited isotope incorporation provides low information content but increased error with respect to model selection.  We have expanded our discussion on this as follows:

“Furthermore, we found that incorporation of less informative datasets can hinder successful model selection, since they lack appreciable signal representing the incorporation of isotopes simulated by moiety models but with the full amount of error of the measured isotopologues.  This lowers the overall isotope incorporation signal to noise ratio, which can lead to increased error in model selection.”

Issue 6:

Please elaborate on section 4.3 (Optimization methods) for a more general audience.

Response:

We have expanded on this section as follows:

"The SAGA-optimize method is a combination of simulated annealing and genetic algorithm optimization methods that utilizes the population and crossover concepts from genetic algorithm to improve the optimization speed and consistency over older more traditional implementations of simulated annealing, allowing SAGA-optimize to produce better quality optimization results more efficiently (i.e. with fewer overall number of optimization steps). The ‘TNC’ method is designed for optimizing non-linear functions with large numbers of independent variables[16]. The SLSQP method uses Sequential Least Squares Programming, which is an iterative method for constrained nonlinear optimization[17]. ‘L-BFGS-B’ is a limited-memory algorithm for solving large nonlinear optimization problems subject to simple bounds on the variables[18].”

Issue 7:

Why are there two sections named “Optimization methods”?

Response:

Oops!  Section 4.4 is supposed to be Model Selection Estimators.  We have fixed this.

Reviewer 2 Report

In this study, the authors studied the factors which can affect the model selection for metabolic flux analysis. They found over-fitting could lead the model selection failure, but combined the multiple datasets could reduce this effect. Some comments below:

  • If possible, could the authors provide the link for downloading the software in manuscript? It will be more convenient for other readers to evaluate the software.
  • Could the authors provide more details in experimental part, especially the cell culture part? Did the authors use 100% U-13C-glucose as isotope tracer? In metabolic flux analysis, 20% glucose + 80% U-13C-glucose is a more common choice. Could the authors explain why they choose 100% U-13C-glucose as tracer? Did the authors get a chance to test other isotope tracers? (e.g., 1,2-13C-glucose, 1,6-13C-glucose are well-known tracers for more accurate flux analysis).

Author Response

Reviewer 2:

In this study, the authors studied the factors which can affect the model selection for metabolic flux analysis. They found over-fitting could lead the model selection failure, but combined the multiple datasets could reduce this effect. Some comments below:

Issue 1:

If possible, could the authors provide the link for downloading the software in manuscript? It will be more convenient for other readers to evaluate the software.

Response:

This was an oversight on our part when we reformatted for this journal, since the older preprint on bioRxiv had all of these links.   We have now added links to the software on Github and to the FigShare repository with all of the data, results, and the scripts used to generate the results.

“Computer Code and Software:

The moiety_modeling and SAGA-optimize packages are available on:

GitHub - https://github.com/MoseleyBioinformaticsLab/moiety_modeling, https://github.com/MoseleyBioinformaticsLab/SAGA_optimize.

PyPI - https://pypi.org/project/moiety-modeling/ , https://pypi.org/project/SAGA-optimize/ .

Project documentation is available online at ReadTheDocs: https://moiety-modeling.readthedocs.io/en/latest/, https://saga-optimize.readthedocs.io/en/latest/.

Data Availability: All data used and the results generated in this manuscript are available on figshare: https://figshare.com/articles/moiety_model_selection/10279688”

Issue 2:

Could the authors provide more details in experimental part, especially the cell culture part? Did the authors use 100% U-13C-glucose as isotope tracer? In metabolic flux analysis, 20% glucose + 80% U-13C-glucose is a more common choice. Could the authors explain why they choose 100% U-13C-glucose as tracer? Did the authors get a chance to test other isotope tracers? (e.g., 1,2-13C-glucose, 1,6-13C-glucose are well-known tracers for more accurate flux analysis).

Response:

We did not generate these datasets and they have been discussed in prior publications.  We have emphasized using these references for further details about their experimental design and the rationale for their experimental design as follows:

“Look to the reference associated with each dataset for more details on their experimental design, implementation, and rationale.”

Round 2

Reviewer 1 Report

The reviewer's comments from the previous round have been addressed.